# Optimization of Electrospray Deposition Conditions of ZnO Thin Films for Ammonia Sensing

**DOI:** 10.3390/nano14121008

**Published:** 2024-06-11

**Authors:** Georgi Marinov, Gergana Alexieva, Katerina Lazarova, Rositsa Gergova, Petar Ivanov, Tsvetanka Babeva

**Affiliations:** 1Institute of Optical Materials and Technologies “Acad. J. Malinowski”, Bulgarian Academy of Sciences, Akad. G. Bonchev Str., bl. 109, 1113 Sofia, Bulgaria; gmarinov@iomt.bas.bg (G.M.); klazarova@iomt.bas.bg (K.L.); petar@iomt.bas.bg (P.I.); 2Faculty of Physics, University of Sofia, 5 James Bourchier Blvd., 1164 Sofia, Bulgaria; 3Central Laboratory of Solar Energy and New Energy Sources, Bulgarian Academy of Sciences, 72 Tsarigradsko Chaussee, 1784 Sofia, Bulgaria; rositsa.gergova@gmail.com

**Keywords:** ZnO, electrospray deposition, QCM sensing, ammonia sensing

## Abstract

This study focuses on the influence of electrospray deposition parameters on the morphology, topography, optical and sensing properties of ZnO films deposited on gold electrodes of quartz crystal resonators. The substrate temperature, precursor feed rate and emitter’s voltage were varied. Zinc acetate dehydrate dissolved in a mixture of deionized water, ethanol and acetic acid was used as a precursor. The surface morphology and average roughness of the films were studied by scanning electron microscopy (SEM) and 3D optical profilometry, respectively, while the optical properties were investigated by diffuse reflectance and photoluminescence measurements. The sensing response toward ammonia was tested and verified by the quartz crystal microbalance (QCM) method. The studies demonstrated that electrospray deposition parameters strongly influence the surface morphology, roughness and gas sensing properties of the films. The deposition parameters were optimized in order for the highest sensitivity toward ammonia to be achieved. The successful implementation of the electrospray method as a simple, versatile and low-cost method for deposition of ammonia-sensitive and selective ZnO films used as a sensing medium in QCM sensors was demonstrated and discussed.

## 1. Introduction

The increasing rates of the environmental pollution from different industrial sources and automobiles and the consequent deterioration in the quality of life requires enhanced control of the release of hazardous chemicals. The synthesis of new materials, the modification of existing ones’ properties and their appropriate integration with transducers operating on various physical principles are remarkably consistent with the development and refinement of precise environment monitoring systems for gas sensing. The implementation of low-cost and flexible physical transducers along with energy-efficient and ecologically friendly material synthesis and deposition methods is a crucial part in the development of sensing devices.

Metal oxide nano-materials are widely studied owing to their diversity in surface morphology, high sensing performance and relatively easy and controllable synthesis [1]. Due to its unique properties [2], like transparency in the VIS and NIR spectral ranges, low resistivity, good thermal stability and biocompatibility, ZnO is traditional but increasingly relevant material with applications in different fields such as optoelectronics, pharmacy, and bio- and gas sensing. ZnO nanostructures have been widely used in light emitters, field-effect transistors, lasers, sensors and solar cells [3,4,5,6,7]; for color sensing [8]; and as transparent conductive oxide [9]. The morphology-controlled synthesis of ZnO nanostructures has been extensively explored; various nanostructures like nanotubes, nanorods, nanoneedles, nanohelixes and nanodisks have been synthesized by adjusting preparation methods and conditions [3,10,11,12,13,14,15]. Among the most common methods applied for the synthesis of ZnO nanostructures with various morphologies, sizes and sensing properties are chemical vapor deposition [16,17], atomic layer deposition (ALD) [18,19,20], magnetron sputtering [21,22,23,24], pulsed laser deposition (PLD) [25], electrochemical deposition [26], sol–gel [27], electron beam evaporation [28], carbothermal transport growth [29], etc.

Recently, we used the electrospray method for the deposition of ZnO with well-controlled properties [30,31,32]. The method belongs to the family of the so-called “wet” deposition methods because it utilizes solution processing. It is cost-effective, environmentally friendly and requires relatively simple and inexpensive equipment. Moreover, electrospray methods are quite flexible and offer easy control on properties of the produced structures by varying the deposition parameters. Furthermore, the deposition process is performed at atmospheric pressure and moderate temperatures and the precursors used are inexpensive and easily available. 

Most of the reported gas sensors utilizing ZnO as a sensing material are based on conductance/resistance changes caused by the chemisorption process of reactive gases on the surface of ZnO [3]. In the present work, we used the highly sensitive quartz crystal microbalance (QCM) method for direct mass measurement. This method has been widely explored for sensing purposes due to its high precision, accuracy and simplicity. The working principle is based on resonance frequency change produced from the mass loading on the active layer (ZnO thin film in our case) due to gas adsorption [33,34]. The sensing ability of the films at room temperature was tested towards ammonia vapors. As known, ammonia is utilized extensively in many industries. Detection of ammonia is especially important as it could lead to toxic build-up in tissues and blood [35]. The toxicity is related to the concentration in the environment and the exposure time. Ammonia poisoning occurs mainly through the respiratory tract [36]. Its room temperature detection at different concentration levels still remains a challenging task [37].

In this paper, we studied the impact of electrospray deposition parameters on the morphological, optical and gas sensing properties of ZnO films deposited on quartz resonators. This research is focused on the optimization of electrospray deposition parameters (emitter’s voltage, substrate temperature, and precursor feed rate) in order to achieve the highest sensitivity of the resulted structures toward ammonia and to compare their sensing abilities. Without claiming to be exhaustive on the subject, we aimed at highlighting some aspects of electrospray deposition techniques in relation to its use in gas sensing devices relying on the acoustic principle of analyte signal conversion.

## 2. Materials and Methods

### 2.1. Deposition of ZnO Thin Films

The electrospray deposition of ZnO films was performed in a homemade vertical setup, presented in Figure 1. The precursor solution was placed in the syringe (2) connected to a syringe pump (1), thus enabling constant and controllable delivery of the solution to the emitter with feed rates of 10 and 15 µL/min. For the emitter (3), a stainless steel needle of outer to inner diameter of 508 to 241 microns, respectively, was used. The sample—a quartz crystal resonator (4)—was placed on the collector (5), a stainless steel–duralumin plate grounded safely and heated at temperatures in the range 150–200 °C by a heater (6). The emitter-to-collector distance was kept constant at 60 mm and a high voltage (15 kV, 18 kV) with positive polarity was applied via a DC power supply (7) (Applied Kilovolts, Worthing, UK). The deposition was started when the working temperature was established and lasted 60 min for all samples, thus ensuring similar film thicknesses of 200 nm. Table 1 presents the exact parameters of the electrospray deposition process for each of the prepared samples.

The ZnO precursor for the electrospraying was prepared by dissolving 0.400 g of zinc acetate dehydrate in 1.8 mL deionized water, followed by dilution in ethanol (12 mL) and addition of few acetic acid drops for solution clearing. The solution was stirred for 2 h at room temperature and aged for 24 h under ambient conditions. A sol–gel process was involved in generation of ZnO from the precursor consisting of hydrolysis/alcoholysis and condensation. More details about the reactions involved can be found elsewhere [38].

### 2.2. Study of Films’ Properties

A scanning electron microscope (Philips 515, Philips, Eindhoven, The Netherlands) was used for examination of surface morphology of ZnO thin films. The SEM pictures were taken at 8 kV accelerating voltage at a working distance in the range from 6 to 8 mm. The surface topography and roughness of the films over a large area (6750 squared microns) were studied by optical profiler (Zeta-20, Zeta Instruments, Gallatin Valley, MT, USA). The surface roughness measurements were made at five different spots on the sample, each with an area of 6750 squared microns, and their root mean squared average value (*Sq*) was calculated. 

Wettability properties were studied by measuring the water contact angle (WCA) geometrically, determined by a liquid at the three phases’ boundary intersection between the solid, liquid and ambient gas (air). All measurements were performed at room temperature and a relative humidity (RH) of 55% with a DSA30 Drop Shape Analyzer (KRÜSS GmbH, Hamburg, Germany) by using 6 µL of distilled water for each sample dispensed on the surface with a rate of 2.67 µL/s. For each WCA measurement, multiple readings (30–50) within 3–5 s were taken 1–2 s after water drop stabilizing. 

Photoluminescence of the films was measured with a FluoroLog 3–22, Horiba Jobin Yvon spectrofluorometer (Glasgow, UK) at an excitation wavelength of 325 nm and an emission range of 350 nm to 800 nm. Other parameters of the spectral measurements were slit width of 5 nm and step of 1 nm. 

Diffuse reflectance was estimated by an UV-VIS-NIR Shimadzu UV 3600 spectrophotometer (Shimadzu Scientific Instruments, Columbia, MD, USA) equipped with an integrating sphere with a diameter of 60 mm.

### 2.3. Study of Gas Sensing Properties

The study on gas sensing properties of ZnO films was performed using the gravimetric principle of the quartz crystal microbalance (QCM) method. For this purpose, one of the electrodes of AT quartz resonators (Figure 2) was coated with ZnO film that is expected to react specifically to sorption of gas molecules. As known [33,34], from the surface loading, a change in the resonant frequency occurs. Thus, the sensitivity and affinity towards gas analyte can be derived from the selective layer’s behavior deposited on the resonant plate. The selected AT-cut quartz resonators work on the basis of thickness shear oscillations and possess satisfactory temperature stability and a good quality factor. The resonators were purchased from the company “Novaetech”, Italy. They were designed to operate on fundamental resonance frequency of 10 MHz. The quartz plate was metalized on both sides with Au electrodes with diameters of 12.2 and 6.4 mm, respectively, and the ZnO layer was deposited onto a bigger one. The diameter of the quartz plate was 14.0 mm. 

The quartz resonator covered with ZnO film was placed on a holder at a vertical position in a test Plexiglas chamber with a volume of 22.5 L (Figure 2). The size of a chamber is selected such that its volume is very close to the volume of 1 mol of ideal gas at room temperature and atmospheric pressure (22.4 l). The chamber temperature and the resonator’s oscillation frequency were monitored in time using QCM Arduino Shield (“Novaetech”, Italy) connected to a computer. The temperature during the measurements was kept constant (at 25 °C) and controlled within 1 deg. The measurement setup is illustrated in Figure 2.

Different concentrations of studied vapors were established by sequential injection in the chamber of constant volume of liquid (0.1 mL) through hermetic inlet. The resulting air–gas mixture was constantly homogenized by fan in the course of the measurements. The off option after each sensing experiment was performed by opening the top wall of the chamber and exposing the resonator to ambient atmosphere. Each subsequent drop of the liquid was introduced at the end of a relaxation interval when a state close to equilibrium was reached. The gas concentrations were converted in ppm units using the procedure described in [39]. The volume evaporated by a given amount of the analyte was normalized to the volume of the chamber where the volumes of accessories—the oscillator and fan—were excluded. The well-known fact that 1 mol of ideal gas has a volume of 22.4 l at room temperature and atmospheric pressure was taken into account.

## 3. Results and Discussion

SEM pictures of the surface morphology for the films deposited on Au electrode of quartz crystal resonator are presented in Figure 3a–e. Figure 3f presents the column bar plot of surface roughness values of the films measured by three-dimensional (3D) optical profiler. The values are presented in Table 2 as well. It can be seen that different deposition parameters lead to different morphology of the films, which was expected. All films are continuous, most of them possess porous structures with relatively large air gaps and agglomerates of different sizes scattered on them (Figure 3a–e). Generally, the decrease in emitter voltage from 18 kV to 15 kV at a constant feed rate and substrate temperature (R2Z2 vs. R2Z4) can be associated with size reduction in the agglomerates (Figure 3b vs. Figure 3d), while larger and relief-like aggregates are formed when the solution feed rate is increased from 10 µL/min (R2Z4) to 15 µL/min (R2Z5) (Figure 3d vs. Figure 3e, respectively) reaching the roughest surface. 

It is seen from Figure 3f and Table 2 that roughness values above 200 nm were obtained at a feed rate of 15 µL/min (samples R2Z and R2Z5), while at a lower feed rate (10 µL/min), smoother films were deposited. If the voltage is kept at 18 kV, the roughness of the deposited at 10 µL/min film is still substantially high (*Sq* = 158 nm for R2Z2 film); however, when the voltage is decreased to 15 kV, the roughness drops significantly: from 158 nm for R2Z2 film to 65 nm and 41 nm for R2Z4 and R2Z3 films, respectively. The smoothest film (R2Z3) is obtained at the highest substrate temperature (200 °C). We should note here that additional measurement of the surface of the bare resonator indicated that its roughness is less than 20 nm, which means that the roughness of films is not affected by the roughness of the bare resonators.

Three-dimensional surface images obtained by three-dimensional (3D) optical profilometry were used to track the quality of the films: presence/absence of cracks, defects, artifacts, etc. Figure 4a,b demonstrate the surface of samples R2Z3 and R2Z5 selected as the smoothest (*S_q_* = 41 nm) and the roughest (*S_q_* = 215 nm) thin film samples. The difference in surface topography is very well distinguished. R2Z5 surface consist of huge number of pikes with different height up to 1–1.5 microns, while the surface of R2Z3 is very smooth and homogenous.

Figure 4c presents the diffuse reflectance, *R_d_*, of R2Z5 and R2Z3 thin films. As can be expected considering the rough surface of R2Z5, it has significantly stronger diffuse reflectance as compared to R2Z3. In the visible and NIR spectral ranges (wavelength longer than 400 nm), where ZnO is transparent, *R_d_* for both samples decreases with wavelength but to different extents: from 20.4% at 390 nm to 12.5% at 1000 nm for R2Z5 and from 6.6% at 390 nm to 0.9% at 1000 nm for R2Z3. The steep drop of *R_d_* at wavelengths less than 360 nm is associated with high light absorption in the films in the UV spectral range that prevents scattering in the volume. Absorption edge for both films is well distinguished in the spectral range 360–390 nm. 

The measured diffuse reflectance values *R_d_* were used for calculation of optical band gap, *Eg*, according to the theory of P. Kubelka and F. Munk [40]. Generally, *Eg* for ZnO was calculated using the Tauc plot [41], assuming direct transition between valence and conduction bands [30,31]. This means that (α**E*)^2^, where α is the absorption coefficient in cm^−1^ and *E* is the light energy in eV, is plotted versus *E.* The value of optical band gap, *Eg*, is calculated from the cross point of the linear approximation of the plot with x-axis at y-axis value of zero. We should note here that the power degree of the product (α**E*) is 2 because of the assumption of direct transitions in ZnO. In our case, instead of absorption coefficient α we used Kubelka–Munk function *F*(*R*) [40] that is described with Equation (1):(1)$$F\left(R\right)=\frac{{\left(1-R\right)}^{2}}{2R}$$

Then, similarly to the case described in detail above, the function (*F*(*R*)**E*)^2^ is plotted versus light energy *E* (Figure 4d) and the optical band gap *Eg* is determined from the linear part of the plot at (*F*(*R*)**E*)^2^ = 0, as illustrated in Figure 4d. Assuming direct transitions, the calculated values for *Eg* were 3.3 ± 0.1 eV for both R2Z5 and R2Z3 thin films. These values are in very good agreement with the value of 3.29 eV previously obtained for thin transparent ZnO films with roughness of 26 nm deposited on silicon substrate at substrate temperature of 300 °C, feed rate of 15 μL/min and voltage of 18 kV [30]. Therefore, a conclusion could be drawn that the experimental deposition parameters of electrospray do not significantly influence the optical band gap of thin ZnO films.

The next step in our investigation concerns the sensing properties of the films. Figure 5 shows the change in frequency Δ*f* for the studied ZnO thin films when exposed to ammonia vapors of different concentrations. Ammonia vapors are absorbed on the surface and in the volume of the films, increasing the mass deposited on the resonator and thus reducing its oscillating frequency. This is the reason for the negative Δ*f* values. It is expected that Δ*f* increases with vapor concentration due to the increase in mass of the ZnO films. From the slope of the linear part of the plot Δ*f* versus concentration, the sensitivity of the sensor (in Hz per ppm) is determined. The calculated sensitivity is presented in Table 2.

It is seen from Figure 5 and Table 2 that R2Z2 film deposited at a substrate temperature of 150 °C, a voltage of 18 kV and a feed rate of 10 μL/min has the highest sensitivity (0.12 Hz/ppm). The lowest sensitivity (0.02 Hz/ppm) was obtained for sample R2Z3, which was deposited at the highest substrate temperature (200 °C). As known, the sensing mechanism of ZnO nanostructured materials is based on the adsorption process of oxygen molecules on their surface in the presence of atmospheric air [42]. When ZnO films are exposed to NH_3_ vapors, a surface reaction takes place. As a consequence of the high electronegativity of the oxygen molecules and electrons extracted from the conduction band of the ZnO layer, a formation of oxygen ions occurs [42]. As a result, from the interaction of NH_3_ molecules with the sorbed oxygen ions, the following reaction takes place [42,43]:(2)$$4{\mathrm{NH}}_{3}+3O_{2}^{-}\left(\mathrm{ads}\right)\to 2N_{2}+6H_{2}O+3e^{-}$$

The films with a more developed surface contain more centers for gas adsorption and a higher frequency change is expected. Their sensitivity is expected to be greater because of their higher probability for interaction with gas molecules. However, as can be seen from Table 2, the roughest layers (R2Z and R2Z5 with *Sq* more than 200 nm) do not have the highest sensitivity and the layer with the highest sensitivity (R2Z2) does not possess the most developed surface (*Sq* = 158 nm). It is also interesting to note that films R2Z4 and R2Z5, which have similar sensitivity, have very different surface roughness, 65 nm and 215 nm, respectively. Therefore, in the ammonia vapor case, the developed surface is not the only factor that determines sensitivity.

The other factor that should be considered in this study is the presence of defects in the ZnO crystal lattice, which are expected to increase the absorption of oxygen molecules, thus enhancing the sensitivity. In order to evaluate the presence of such defects, we measured the photoluminescence of the films under excitation at a wavelength of 325 nm. The measured spectra are presented in Figure 6, and for convenience, the ammonia vapor sensitivity values, *S*, in Hz/ppm, are indicated in the figure as well. Strong photoluminescence in the visible range is seen for the R2Z, R2Z2 and R2Z5 films, indicating defects in the crystal lattice and stronger disorder as compared to the R2Z3 and R2Z4 films. We should note that a correlation is observed between the roughness of the layers and their emission in the visible region: rougher layers exhibit stronger photoluminescence than smoother ones.

The strong photoluminescence of the R2Z, R2Z2 and R2Z5 layers in the visible range of the spectrum can be attributed to the presence of defects in them, and we can expect high sensitivity when they are exposed to vapors. It can be seen from Figure 6 that for the R2Z2 layer, these expectations are justified (the sensitivity is 0.12 Hz/ppm), but for the R2Z and R2Z5 samples, the sensitivity is several times lower, despite the presence of defects.

The results of the studies of the morphology, roughness and defects in the layers show that there are additional factors that influence the sensor sensitivity. In order to investigate the impact of the hydrophilic–hydrophobic balance of the surface, contact angle measurements of a water droplet on the surface of the layers were made, and the results are listed in Table 2. As can be seen from Table 2, the WCA values are significantly impacted by the deposition conditions, even approaching in case of sample R2Z superhydrophobic state. The results show that for the two roughest layers, R2Z and R2Z5, the water contact angles are 146° and 136°, i.e., these layers are highly hydrophobic. Two factors are essential for the wettability of the surfaces—their polarity and the trapped air pockets within a textured surface that leads to a notable increase in the contact angle [44]. In contrast, the R2Z2 layer has a water contact angle of 65°, i.e., it is hydrophilic. 

It can be concluded that due to the hydrophobic character of their surface, the R2Z and R2Z5 layers repel ammonia molecules more strongly than the R2Z2 layer, which is hydrophilic. This is the possible reason for their lower sensitivity compared to R2Z2, despite their more developed surface. As for the R2Z2 layer, its high sensitivity is due to the rough but hydrophilic surface and the presence of defects.

The main results of our study are summarized in Figure 7, which demonstrates the normalized values of roughness (*Sq*), photoluminescence (PL) values at peak wavelength, water contact angles (WCAs) and sensitivity (*S*) for all studied samples. We have demonstrated that there are three main factors that governed the sensitivity of the electrosprayed ZnO films towards ammonia vapors: (i) a highly developed surface, which we associated with high roughness, *Sq*; (ii) lattice defects, meaning high photoluminescence, PL, in the visible range and (iii) small water contact angle, WCA, indicating a hydrophilic surface. The sample that satisfies all three criteria is R2Z2, and this explains why R2Z2 has the highest sensitivity. The other samples satisfy one or two criteria but none of them fulfill all three. For example, R2Z and R2Z5 have highly developed surfaces and strong PL but are highly hydrophobic (i.e., high WCA), and the latter results in low sensitivity. On the contrary, R2Z3 and R2Z4 have a low WCA (i.e., they are hydrophilic) but their roughness and defect levels are very low.

We should note here that the sensing of ammonia with electrosprayed thin films is highly selective. Our additional measurements revealed that sensitivity in the case of ethanol, for example, is 0.0085 Hz/ppm, which is 14 times smaller as compared to the sensitivity for ammonia.

Comparison with the literature data is very difficult and controversial because the sensitivity depends on experimental parameters such as the thickness of the layer, the operating frequency, the surface area of the resonator, etc., and information about these parameters is very often omitted in the literature. In particular, using the same sensing setup, the sensitivity achieved by the electrosprayed ZnO films is an order of magnitude higher compared to the electrochemically deposited ZnO films [26].

In general, the typical sensitivity values for this type of detection are in the range 0.1–0.5 Hz/ppm when using ZnO. Recently, Zhou and co-authors [45] reported a sensitivity of 4.54 Hz/ppm for ammonia vapor detection, but they used ZnO nanofibers and nanostructures modified with cellulose acetate and polyaniline with thicknesses/mass that significantly exceed the thicknesses/mass of the layers of our work (layer thickness in our case is 200 nm). 

We should note here that the sensing behavior estimated using the QCM method is largely determined by the homogeneity and continuity of the sensitive film that covers the gold electrode of the resonator and the adhesion of the film to the electrode surface. When the continuity of the film is impaired with cavities and pores, the distortion in the film may cause high acoustic losses, strongly worsening the stability and, respectively, the sensing behavior of the device. Fortunately, it was revealed that the electrospray method presented here for deposition of sensitive ZnO films is beneficial from the view point of the above concerns because, due to annealed substrate and charged particles that reached the electrode surface, the covering is very homogeneous, and the ZnO films adhered well on the Au electrode surface. Further efforts will be devoted to enhancing the obtained sensitivity through doping of ZnO thin films with carefully selected dopants.

## 4. Conclusions

The successful implementation of the electrospray method as a simple, versatile and low-cost method for the deposition of ammonia-sensitive and selective ZnO films was demonstrated. The properties of electrosprayed ZnO films were tuned and optimized for application as a QCM sensor for ammonia, and a sensitivity of 0.12 Hz/ppm was achieved using thin films with an approximate thickness of 200 nm. Films deposited at a substrate temperature of 150 °C, a voltage of 18 kV and a feed rate of 10 μL/min exhibited the highest sensitivity (0.12 Hz/ppm). The lowest sensitivity (0.02 Hz/ppm) was obtained for samples deposited at substrate temperature of 200 °C, a voltage of 15 kV and a feed rate of 10 μL/min. It was demonstrated that for achieving the highest sensitivity, the thin films should simultaneously satisfy the following three criteria: (i) a highly developed surface, which we associated with high roughness; (ii) the presence of lattice defects, indicating high photoluminescence in the visible range and (iii) a small water contact angle, evidencing a hydrophilic surface.

Considering the stable mechanical and chemical properties of the electrosprayed ZnO films as well as their relatively low cost of production, future efforts will be dedicated to enhancing their sensitivity through doping and structuring in order to find applications for gas and bio-sensing.

## Figures and Tables

**Figure 1 nanomaterials-14-01008-f001:**
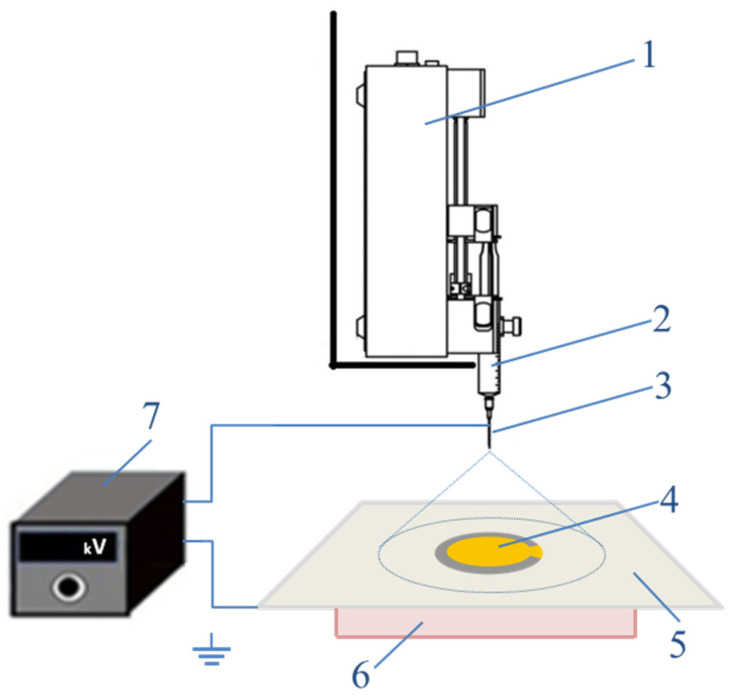
Experimental setup for electrospray deposition: syringe pump (1), syringe with ZnO precursor (2), emitter (3), quartz resonator (4), collector (5), heater (6), high-voltage supplier (7).

**Figure 2 nanomaterials-14-01008-f002:**
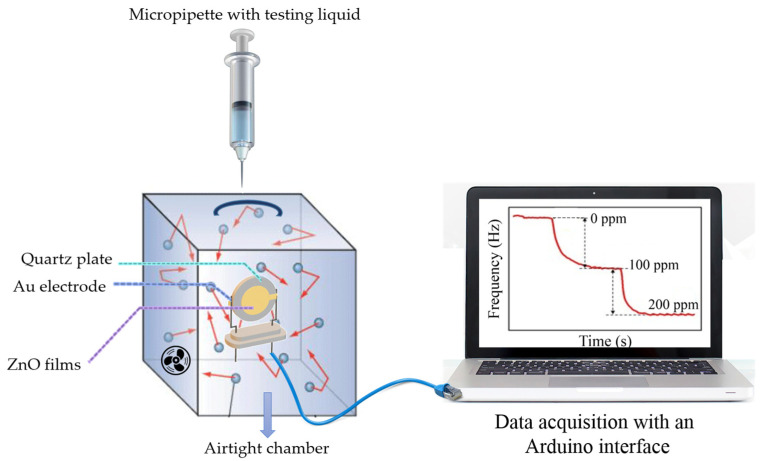
Experimental setup for gas sensing measurements.

**Figure 3 nanomaterials-14-01008-f003:**
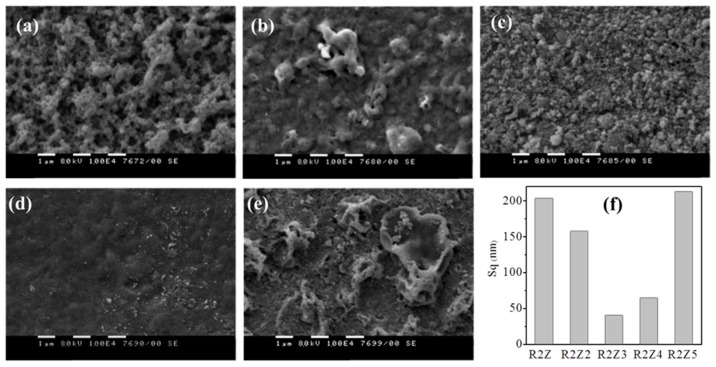
SEM images of ZnO films obtained by electrospray method at different deposition parameters, respectively: (**a**) R2Z; (**b**) R2Z2; (**c**) R2Z3; (**d**) R2Z4; (**e**) R2Z5; (**f**) surface roughness *Sq* in nm.

**Figure 4 nanomaterials-14-01008-f004:**
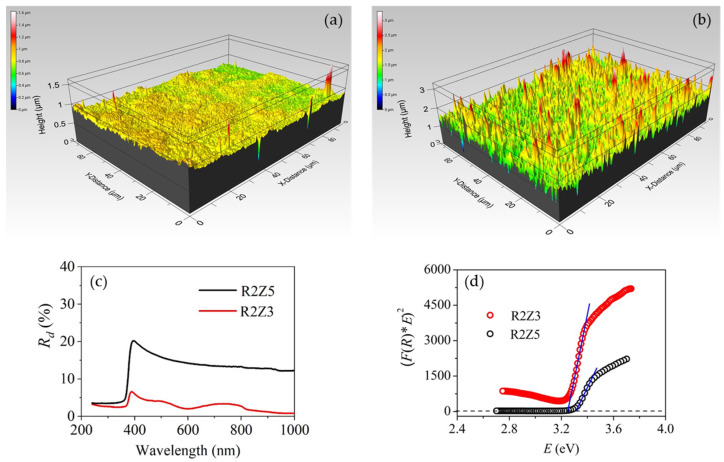
Three-dimensional images of the surface of R2Z3 (**a**) and R2Z5 (**b**) thin films obtained by electrospray at different deposition parameters denoted in Table 1, diffuse reflectance spectra of both films (**c**) and Kubelka–Munk determination of optical band gap energy (*Eg*) (**d**).

**Figure 5 nanomaterials-14-01008-f005:**
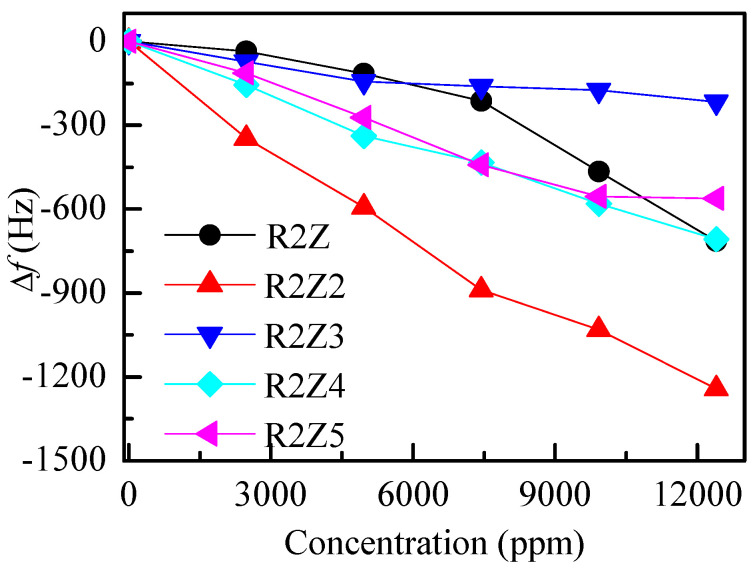
Frequency changes in quartz resonators with ZnO films electrosprayed on them in response to varying amounts of ammonia vapor. The sample codes are listed in Table 1.

**Figure 6 nanomaterials-14-01008-f006:**
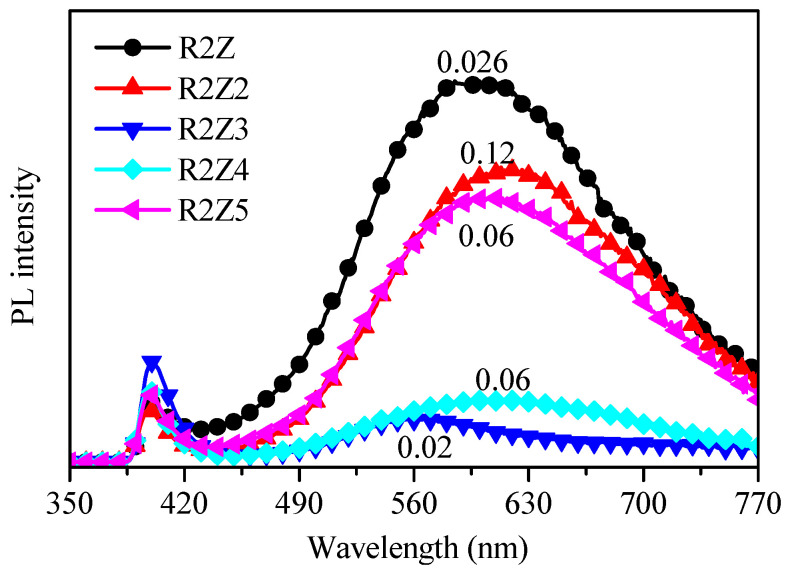
Photoluminescence (PL) spectra of ZnO thin films deposited by electrospray method at different experimental conditions listed in Table 1. The number close to each spectrum indicates the sensitivity of the film in Hz/ppm when exposed to ammonia vapors.

**Figure 7 nanomaterials-14-01008-f007:**
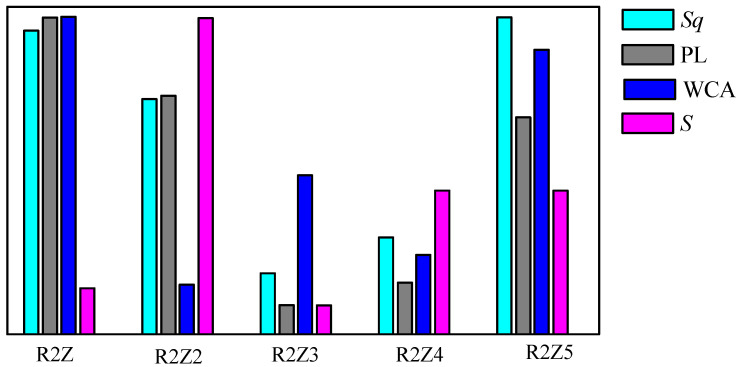
Normalized surface roughness, *Sq* (cyan); photoluminescence, PL (grey); water contact angle, WCA (blue); and sensitivity, S (magenta) of studied thin ZnO films with sample codes indicated in Table 1.

**Table 1 nanomaterials-14-01008-t001:** Sample codes, substrate temperature, applied voltage and feed rate for electrospray deposition of ZnO thin films.

Sample Code	Substrate Temperature (°C)	Voltage (kV)	Feed Rate (µL/min)
R2Z	170	18	15
R2Z2	150	18	10
R2Z3	200	15	10
R2Z4	150	15	10
R2Z5	150	15	15

**Table 2 nanomaterials-14-01008-t002:** Sample code and surface roughness: *Sq*; sensitivity: *S*; water contact angle: WCA; with corresponding errors.

Sample Code	*S_q_* (nm)	*S* (Hz/ppm)	WCA (deg)
R2Z	204 ± 5	0.026 ± 0.002	146 ± 0.9
R2Z2	158 ± 4	0.12 ± 0.004	65 ± 0.1
R2Z3	41 ± 2	0.02 ± 0.002	98 ± 0.6
R2Z4	65 ± 3	0.06 ± 0.001	74 ± 1.2
R2Z5	215 ± 5	0.06 ± 0.001	136 ± 0.7

## Data Availability

Data is contained within the article.

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
