# Peer review of "Optimization of Electrospray Deposition Conditions of ZnO Thin Films for Ammonia Sensing"

_nanomaterials, 2024, doi:10.3390/nano14121008_

Round 1

Reviewer 1 Report

Comments and Suggestions for Authors

This article studies the morphology, the optical and sensing properties of ZnO films deposited on gold electrodes of quartz crystal resonators by means of electrospray deposition. In particular, it investigates the influence of electrospray deposition parameters on the surface morphology, roughness and gas sensing properties of the films tested with ammonia.   

The manuscript needs to be revised and the following main points need to be carefully addressed before it can be considered for publication: 

1.     Due to the large literature in the field, the novelty of the paper should be clearly addressed in the manuscript. 

2.     The introduction needs to be improved. For sake of completeness, among the other applications of ZnO (lines 40-41), at least a relevant example of the important application of ZnO as transparent conductive oxide, such as this one [https://doi.org/10.1016/j.apsusc.2014.05.225], and for the colorimetric-sensing [https://doi.org/10.1155/2023/3603680] should be reported. 

3.     Moreover, among the other synthesis methods used for the synthesis of ZnO nanostructures already reported at lines 46-48, it is indeed important to mention at least the carbo-thermal transport growth [DOI:10.1007/s00339-007-3946-4] and the electron beam evaporation [DOI10.1016/j.mssp.2017.08.015]. 

4.     In the paragraph “2.1 Deposition of ZnO thin films” the chemical reactions for the material preparation should be reported. 

5.     In the paragraph “2.2 Study of films’ properties” the information on the measurement conditions are missing, such as the accelerating voltage and the working distance used during the SEM observations. 

6.     Regarding the PL measurements, the wavelength of the excitation source is reported, but which is the excitation source used? A lamp or a laser? With which resolution?

7.     The values reported in Table 2 should be completed with their errors. 

8.     On line 205, it is important for the reader that the mathematical expression for calculating the Energy Gap (Eg) of the samples used is reported, instead of referring to Ref [34] without providing other information, also because this reference it is quite difficult to find as it is very old (1931).

9.     After showing the way how the Eg was calculated, the values of Eg (line 206) should be reported with their errors.  

10.  By using the electrospray deposition method for synthetizing the structures, how was the degree of control of the synthesis process? How the reproducibility of the samples was? The authors should mention and discuss in detail this aspect in the manuscript. 

11.  Furthermore, the conclusions should highlight the value that the manuscript adds to the current literature in the field and outline some perspectives. 

12.  About Ref. [30], it should be possibly replaced with a newer one reporting the state-of-the-art at date.  

Author Response

Dear reviewer,

thank you very much for your positive and fruitful review. All your recommendations were considered and the manuscript was revised accordingly. Please find as follows our answers to your questions, comments and amendments.

Comment 1: Due to the large literature in the field, the novelty of the paper should be clearly addressed in the manuscript.

Answer: We agree with the reviewer that in the literature there are a huge number of studies devoted to ZnO. It seems that the scientific interest in this material not only does not weaken but it is observed an increase in the number of the studies in recent years. The main reasons are the unique properties of ZnO and its suitability for different modern applications. Although a lot of deposition methods have already been implemented for production of ZnO films and structures, there is still a demand for simple, inexpensive, flexible and eco-friendly methods of deposition. Therefore our main purpose, i.e the novelty of the paper, is to demonstrate the successful implementation of the electrospray method as a simple, versatile and low-cost method for deposition of ammonia sensitive and selective ZnO films. Moreover the deposition takes place in an ambient environment and moderate temperatures without an additional post deposition annealing step and uses inexpensive precursors and eco-friendly solvents. The manuscript is revised accordingly.

Comment 2:  The introduction needs to be improved. For sake of completeness, among the other applications of ZnO (lines 40-41), at least a relevant example of the important application of ZnO as transparent conductive oxide, such as this one [https://doi.org/10.1016/j.apsusc.2014.05.225], and for the colorimetric-sensing [https://doi.org/10.1155/2023/3603680] should be reported.

Answer: Both papers have been included in the introduction.

 Comment 3: Moreover, among the other synthesis methods used for the synthesis of ZnO nanostructures already reported at lines 46-48, it is indeed important to mention at least the carbo-thermal transport growth [DOI:10.1007/s00339-007-3946-4] and the electron beam evaporation [DOI10.1016/j.mssp.2017.08.015].

Answer: Both papers have been included in the introduction.

Comment 4:  In the paragraph “2.1 Deposition of ZnO thin films” the chemical reactions for the material preparation should be reported.

Answer: The chemical reactions involved in the preparation of ZnO are hydrolysis / alcoholysis and condensation as part of the sol-gel reactions. According to the reviewer recommendations a new reference was added where all reactions are very well described and visualized.

Comment 5: In the paragraph “2.2 Study of films’ properties” the information on the measurement conditions are missing, such as the accelerating voltage and the working distance used during the SEM observations.

Answer: The accelerating voltage is 8 kV and can be seen from the pictures in Figure 3. The working distance is varying between 6 and 8 mm. These details were added in the revised version of the manuscript in section 2.2 Study of films’ properties.

Comment 6: Regarding the PL measurements, the wavelength of the excitation source is reported, but which is the excitation source used? A lamp or a laser? With which resolution?

Answer: For PL measurements we used a commercially available high sensitivity spectrofluorometer FluoroLog 3-22 produced by Horiba Jobin Yvon. It is a fully automated modular system allowing measurement of light emission of practically any type of samples, including very thin layers with a very low emission as well as quantum dots. The excitation source of FluoroLog 3-22 is a high power (450 W) xenon lamp. Additionally it is equipped with double-grating monochromators in excitation and emission in the range 200-950nm with wavelength accuracy of 0.5 nm.

Comment 7. The values reported in Table 2 should be completed with their errors.

Answer: We are grateful to the reviewer for pointing out this lack. In the revised version we have added the uncertainties of all parameters listed in Table 2.

Comment 8: On line 205, it is important for the reader that the mathematical expression for calculating the Energy Gap (Eg) of the samples used is reported, instead of referring to Ref [34] without providing other information, also because this reference it is quite difficult to find as it is very old (1931).

Answer: Generally, optical band gap was calculated using the Tauc plot [Tauc, J.; Menth, A. States in the gap. J. Non-Cryst. Sol. 1972, 8–10, 569–585., Ref 41 in the revised manuscript], assuming direct transition between valence and conduction bands. This means that (alfa*E)^2 is plotted versus light energy E, and from the linear part of the curve, Eg is calculated at (alfa*E)^2 = 0 (alfa is the absorption coefficient in 1/cm, E is the light energy in eV, the power degree is 2 because of the assumption of direct transitions in ZnO). In our case, instead alfa we used Kubelka-Munk function F(R) = (1-R)^2 / (2*R) and plotted (F(R)*E)^2 versus E. Optical band gap Eg is determined from the linear part of the plot at (F(R)*E)^2 = 0.

The manuscript is revised accordingly. An additional figure (figure 4d) was added for illustration of optical band gap calculation using the Kubelka-Munk function F(R).

Comment 9:  After showing the way how the Eg was calculated, the values of Eg (line 206) should be reported with their errors.

Answer: In the revised version of the manuscript the uncertainties were added to the values of optical band gap, Eg.

 Comment 10: By using the electrospray deposition method for synthetizing the structures, how was the degree of control of the synthesis process? How the reproducibility of the samples was? The authors should mention and discuss in detail this aspect in the manuscript.

Answer: As we have already mentioned in the manuscript we used home-made set-up for electrospraying (ES). The set-up was constructed and built in a way to offer versatility, security and also a high degree of parameter control in order films with reproducible properties to be deposited. Our experience in ES deposition has revealed that the most important deposition parameters that strongly influence the properties of the films are the flow rate of the precursor, substrate temperature during deposition and high voltage applied to the emitter. The reproducible and stable flow rates of the precursor were guaranteed by the use of a computer controlled syringe pump. The collector, where substrate was placed, is constructed with a heater located under its entire working area and the temperature control is carried out with a PID controller allowing precise temperature control. In order to achieve reproducible high voltage we used a high-quality DC power supplier. All these design solutions guarantee reproducible deposition of the thin films. Furthermore the reproducible deposition was checked and confirmed by examination of film properties such as structure by XRD, morphology by SEM and TEM, optical properties by ellipsometry and PL measurements, transparency by UV-VIS-NIR spectroscopy etc.

 Comment 11: Furthermore, the conclusions should highlight the value that the manuscript adds to the current literature in the field and outline some perspectives.

Answer: We have thoroughly revised the conclusion according to the recommendations

 Comment 12: About Ref. [30], it should be possibly replaced with a newer one reporting the state-of-the-art at date.

Answer: According to the reviewer recommendation reference 30 was replaced with a newer one (Ref 35 in the revised manuscript).

Reviewer 2 Report

Comments and Suggestions for Authors

The development of materials that can be used for ammonia sensing is an urgent and interesting task. However, this paper leaves an ambiguous impression. The title of the paper “Optimization of electrospray deposition conditions of ZnO thin films for ammonia sensing” is suitable for a purely technological work, but it is not very well suited for a scientific paper.

The general comment to the paper is that the scientific novelty of this paper is not clear, this information is not in the Abstract or Conclusions. In this paper the authors have optimized the parameters to produce ZnO films using electrospray method. Since there is almost no comparison with the results of other authors, it remains unclear whether this paper achieved results comparable or superior to other papers. Since the idea to use ZnO thin films for ammonium detection is not original, it is necessary to show that the proposed method of preparation allows to obtain films more suitable for ammonium detection in comparison with other methods. Otherwise, it is not clear what to use this method for.

In the Introduction, the authors write (see Lines 49-50): “Recently, we have demonstrated that electrospray method could be used for deposition of ZnO with well controlled properties [25-27]”. This phrase creates a false impression, as if this method of obtaining ZnO films is proposed by the authors of this work, which is actually not the case, see, for example, the following works: Yan Huai Ding et al. (2009, https://doi.org/10.4028/www.scientific.net/SSP.155.151); A. Hosseinmardi et al. (2012, https://doi.org/10.1016/j.ceramint.2011.10.031).

In this paper it is shown that the best results are obtained for the film with the lowest roughness (40 nm), see Figure 5. The question arises whether this method of film preparation is optimal for obtaining materials for ammonium detection. What happens if we use another method and obtain films with roughness of 5-10 nm?

Regarding the comparison of their results with data of other authors, the article says (see Lines 302-304): “The comparison with literature data is very difficult and controversial because the sensitivity depends on the thickness of the layer, and information about this parameter is very often omitted in the literature”. However, this comparison should be made, otherwise it remains unclear how effective is the method used by the authors of this work to obtain ZnO films.

Regarding Conclusions we can say the following, instead of giving in this section some conclusions made on the basis of the work done, the authors only state the facts:

1) technological parameters of sample preparation influence its morphology and properties - this conclusion is generally obvious;

2) “rougher layers exhibited stronger photoluminescence”;

3) “Other factor that affected the sensitivity was the presence of defects in the ZnO crystal lattice” - probably, the presence of defects inside the crystal lattice cannot affect the absorption properties much. The absorption will be primarily influenced by the surface. Thus, it must be primarily a matter of defects in the surface layer as well as the size of the surface area. Accordingly, smaller ZnO crystallite sizes result in a larger surface area of the film.

Conclusions does not contain any information about the advantages or disadvantages of the proposed method of obtaining ZnO films for ammonia sensing.

Comments on the Quality of English Language

No serious problems with English.

Author Response

Dear reviewer,

thank you very much for your positive and fruitful review. All your recommendations were considered and the manuscript was revised accordingly. Please find as follows our answers to your questions, comments and amendments.

Comment 1: The development of materials that can be used for ammonia sensing is an urgent and interesting task. However, this paper leaves an ambiguous impression. The title of the paper “Optimization of electrospray deposition conditions of ZnO thin films for ammonia sensing” is suitable for a purely technological work, but it is not very well suited for a scientific paper.

Answer: We are very delighted that the reviewer shares our opinion that sensing ammonia is an essential scientific topic. However we cannot agree that the proposed title “Optimization of electrospray deposition conditions of ZnO thin films for ammonia sensing” is not suitable for scientific paper because “leaves an ambiguous impression”. We have selected the title very carefully and we are pretty confident that it expresses very well the content of the manuscript and that the reader will get straight forward the idea of the paper when he reads the title. The manuscript is focused on electrospray deposition of thin ZnO films aiming at its optimization for ammonia sensing. The selected deposition method is very flexible and films with different morphologies and structures could be obtained. This is the reason why we vary (i.e optimize) the electrospray deposition conditions (precursor flow rate, substrate temperature and emitter voltage) aiming at tuning the properties of ZnO films for application of QCM sensing of ammonia.

 Comment 2: The general comment to the paper is that the scientific novelty of this paper is not clear, this information is not in the Abstract or Conclusions. In this paper the authors have optimized the parameters to produce ZnO films using electrospray method. Since there is almost no comparison with the results of other authors, it remains unclear whether this paper achieved results comparable or superior to other papers. Since the idea to use ZnO thin films for ammonium detection is not original, it is necessary to show that the proposed method of preparation allows to obtain films more suitable for ammonium detection in comparison with other methods. Otherwise, it is not clear what to use this method for.

Answer: We are grateful to the reviewer that he points out the weakness of the Abstract and Conclusion. They are thoroughly revised in the viewpoint of both reviewers' recommendations. Also in the revised manuscript we expanded the comparison and have demonstrated that the electrospray method allows deposition of more suitable films for ammonia detection in comparison with other methods. As far as it is known, most of the reported ammonia sensors utilizing ZnO as sensing material rely on conductance changes caused by the sorption process of the gas on the ZnO surface. In fact, there are studies on ammonia sensors with ZnO nanostructures based on acoustic type signal transducers like bulk (BAW) and surface acoustic wave (SAW) resonators, SAW delay lines [1-5]. Both the operating frequency and active surface area affect the devices’ sensitivity. To enhance the device's sensitivity, several deposition methods and approaches, such as doping, creating composite ZnO films are employed. The “wet” electrodeposition methods are highly appealing because they are cost-effective, environmentally compatible, offering easy control and diversity on morphology and stoichiometry of the produced structures at atmospheric pressure and moderate temperatures. Under the same experimental scheme the achieved in [5] for electrochemically grown layers sensitivity is order of magnitude lower. It is obvious that we report original results which only upgrade on previous achievements for advanced gas sensing in the route of development of ZnO based nanostructured films obtained by other deposition methods. The used by us innovation approach to combine two known methods (electrospray deposition and QCM sensing) for production and characterization of the sensing properties of ZnO cannot be found in the literature so far. Further efforts will be devoted to enhance the obtained under optimal deposition conditions sensitivity creating nano-hybrid ZnO material with doping agents.

  1. https://doi.org/10.1016/j.apsusc.2012.11.028
  2. https://doi.org/10.30970/jps.27.3001
  3. https://doi.org/10.1016/j.snb.2016.07.040
  4. reference 45 in the manuscript: https://doi.org/10.1016/j.snb.2023.134072
  5. reference 26 in the manuscript: https://doi.org/10.3390/coatings12010081

Comment 3: In the Introduction, the authors write (see Lines 49-50): “Recently, we have demonstrated that electrospray method could be used for deposition of ZnO with well controlled properties [25-27]”. This phrase creates a false impression, as if this method of obtaining ZnO films is proposed by the authors of this work, which is actually not the case, see, for example, the following works: Yan Huai Ding et al. (2009, https://doi.org/10.4028/www.scientific.net/SSP.155.151); A. Hosseinmardi et al. (2012, https://doi.org/10.1016/j.ceramint.2011.10.031).

Answer: We did not suppose that our way of expression could create a wrong impression that we are the authors that propose the electrospray as a method for thin film deposition. In order to remove any semantic ambiguity we revised the sentence accordingly.

Comment 4: In this paper it is shown that the best results are obtained for the film with the lowest roughness (40 nm), see Figure 5. The question arises whether this method of film preparation is optimal for obtaining materials for ammonium detection. What happens if we use another method and obtain films with roughness of 5-10 nm?

Answer: We cannot agree with the reviewer that the best results are obtained for the film with the lowest roughness. From figure 5 it is seen that the highest sensitivity is obtained for the R2Z2 sample: it is seen from figure 5 that the change of frequency is the strongest for this sample. Its roughness is 158 nm. Generally we have demonstrated in Figure 7 that only samples that satisfies simultaneously all the three criteria are suitable for sensing of ammonia: i) high developed surface that we have associated with high roughness, Sq; ii) lattice defects that means high photoluminescence PL in the visible range and iii) small water contact angle WCA that means hydrophilic surface. Therefore if you used another method and deposited films with roughness 5-10 nm you would not obtain any sensing response. See for example the very small response of sample R2Z3 that has a roughness of 41 nm.

  Comment 5: Regarding the comparison of their results with data of other authors, the article says (see Lines 302-304): “The comparison with literature data is very difficult and controversial because the sensitivity depends on the thickness of the layer, and information about this parameter is very often omitted in the literature”. However, this comparison should be made, otherwise it remains unclear how effective is the method used by the authors of this work to obtain ZnO films.

Answer: Please, see the answer of comment 2 where we explained in detail the results from comparison and how effective our method is. Anyway, please have in mind that in most cases a compromise between sensitivity and effectiveness of deposition should be made. It would be great if highly sensitive and selective media are developed but that would not make sense if very complicated, expensive and nature contaminating methods are used for deposition. Therefore the novelty of the paper is the successful implementation of the electrospray method as a simple, versatile and low-cost method for deposition of ammonia sensitive and selective ZnO films used as sensing medium in QCM sensors. Moreover the deposition takes place in an ambient environment and moderate temperatures without an additional post deposition annealing step and uses inexpensive precursors and eco-friendly solvents.

Comment 6: Regarding Conclusions we can say the following, instead of giving in this section some conclusions made on the basis of the work done, the authors only state the facts:

1) technological parameters of sample preparation influence its morphology and properties - this conclusion is generally obvious;

2) “rougher layers exhibited stronger photoluminescence”;

3) “Other factor that affected the sensitivity was the presence of defects in the ZnO crystal lattice” - probably, the presence of defects inside the crystal lattice cannot affect the absorption properties much. The absorption will be primarily influenced by the surface. Thus, it must be primarily a matter of defects in the surface layer as well as the size of the surface area. Accordingly, smaller ZnO crystallite sizes result in a larger surface area of the film.

Conclusions does not contain any information about the advantages or disadvantages of the proposed method of obtaining ZnO films for ammonia sensing.

Answer: Conclusion is thoroughly revised in the viewpoint of both reviewers' recommendations.

Round 2

Reviewer 1 Report

Comments and Suggestions for Authors

The authors have correctly addressed all the required issues improving the manuscript satisfactorily. Therefore, it is now suitable for publication.

Author Response

Dear Reviewer,

Thank you very much for your positive descisiion.

Sincerely

Prof. Dr. Tsvetanka Babeva

Reviewer 2 Report

Comments and Suggestions for Authors

The authors responded adequately enough to the comments and made corrections in the article. I believe that the article can be published in the presented form, taking into account the following small remarks.

Line 200. ‘optical bang’

misprint, should be ‘optical band gap’ or ‘optical bandgap’.

Line 233. ‘optical band gap’

Line 240. ‘optical bandgap.’

Decide how you are going to write it and write it the same way throughout the paper.

Line 235. ‘the calculated values for Eg were 3.31 ± 0.1 eV for R2Z5 and 3.26 ± 0.1 eV for 235 R2Z3.’

It makes no sense to write the value of Eg to the nearest hundredth if the error in determining this value is one tenth.

Author Response

Dear Reviewer,   

thank you for the fruitful comments and the positive assesment of our manuscript.

All comments were cinsidered in the revised version.

Sincerely,

Prof. Dr. Tsvetanka Babeva